# Unusual Case of Biliary Peritonitis in a Dog Secondary to a Gastric Perforation

**DOI:** 10.3390/vetsci10060384

**Published:** 2023-06-01

**Authors:** Giovanni Pavone, Barbara Castellucci, Silvia Pavone, Valentina Stefanetti, Chiara Vitolo, Sara Mangiaterra

**Affiliations:** 1Futuravet Veterinary Referral Center, 62029 Tolentino, Italy; giovanni.pavone@futuravet.it (G.P.); barbara.castellucci@futuravet.it (B.C.); chiara.vitolo@hotmail.it (C.V.); 2Istituto Zooprofilattico Sperimentale dell’Umbria e delle Marche ‘Togo Rosati’, Via G. Salvemini, 1, 06126 Perugia, Italy; s.pavone@izsum.it; 3Department of Veterinary Medicine, University of Perugia, 06126 Perugia, Italy; valentina.stefanetti@unipg.it

**Keywords:** biliary peritonitis, gastric perforation, dog, NSAIDs

## Abstract

**Simple Summary:**

Biliary peritonitis is a pathological condition that represents a medical emergency both in human and veterinary medicine. Its causes are damage to the hepatobiliary tract, gallbladder, or duodenal tract. In this case report, biliary peritonitis due to gastric perforation is reported in a Bobtail dog, probably following the administration of nonsteroidal anti-inflammatory drugs (NSAIDs). After an elective surgery, the dog was referred for medical management for inappetence, mental depression, and multiple episodes of gastric vomits with traces of blood. Clinical diagnostic tests showed the presence of biliary peritonitis. Due to worsening clinical conditions, the patient was subjected to euthanasia. Macroscopic examination showed a free brownish abdominal effusion and the presence of perforating ulcer in the stomach pylorus region.

**Abstract:**

Biliary peritonitis is a pathological condition representing a medical emergency with a high risk of mortality. This condition is reported in both human and veterinary medicine following biliary tract rupture, extrahepatic biliary obstructions, gallbladder rupture, trauma, or duodenal perforation. In this report, the first-ever case of biliary peritonitis due to gastric perforation in a Bobtail purebred dog is described, which was probably induced by the administration of nonsteroidal anti-inflammatory drugs (NSAIDs). After an elective splenectomy and castration, the dog was referred to our hospital for medical management for inappetence, mental depression, and multiple episodes of gastric vomits with traces of blood. Clinical diagnostic tests showed the presence of biliary peritonitis. Due to worsening clinical conditions, the patient was subjected to euthanasia. Macroscopic examination showed a free brownish abdominal effusion and the presence of perforating ulcer of the stomach pylorus region.

## 1. Introduction

Biliary peritonitis (BP) is a pathological condition characterized by the presence of bile in the abdominal cavity [1]. This condition is reported in both human and veterinary medicine following damage to the hepato-biliary tract, representing a medical emergency with a high risk of mortality [1]. Bile salts and constituents of the bile are extremely toxic outside the enterohepatic tract causing changes in the permeability of vascular structures and consequently, necrosis [2,3,4]. Causes of BP reported include: biliary tract rupture, extrahepatic biliary obstructions, gallbladder rupture, trauma, and duodenal perforation [1]. To the authors’ knowledge, there are no reports of biliary peritonitis following gastric perforation. The present case report describes a condition of biliary peritonitis due to gastric perforation in a Bobtail purebred dog. 

## 2. Case Presentation

A 10-year-old Bobtail (or Old English Sheepdog) neutered male dog was presented to Futuravet Veterinary Referral Centre (Italy) after splenectomy and castration performed three days before due to the presence of a splenic and testicle neoplasia. Two days after the surgery, the dog showed symptoms of inappetence, mental depression and multiple episodes of gastric vomits with traces of blood. Due to this condition, the dog was referred to our hospital for medical management. Clinical history reported the administration of cefalexin (22 mg/kg per os bid), and meloxicam (0.1 mg/kg per os sid); vital signs were normal with heart rate 132/min, respiratory rate 26/min, and a rectal temperature 38.5 °C [101.3 °F], while buccal mucous membrane was pale pink with capillary refill time (CRT) of 2”. Abdominal palpation evocated mild pain. Systolic pressure measured using the Doppler technique was 110 mmHg. A blood exam performed at the presentation revealed non-regenerative anemia, red cells 3.94 M/μL [RI 5.65–8.87], hematocrit (Hct) 26.0 [RI 37.3–61.7%], hemoglobin (Hbg) 8.6 g/dL [RI 13.1–20.5 g/dL], leukocytosis with neutrophilia 28.67 K/μL and monocytosis 2.09 K/μL (RI 0.16–1.12 K/μL). The patient’s venous gas analysis showed hyperlactatemia that may cause or induce metabolic acidosis with 3.8 mmol/L (RI < 2.5 mmol/L) and a bicarbonate concentration of 15 mmol/L (RI 20–24 mmol/L). On abdominal ultrasound, the gastric wall was thickened and distended by fluid. Mild hyperechoic mesentery was visualized without abdominal effusion and compatible with the surgery performed before. The dog was hospitalized and was supported with intravenous (IV) fluids. To reduce gastric repletion, a nasogastric feeding tube was placed, and 356 mL of brown gastric content was aspirated. Suspecting a gastric ulceration, anti-inflammatory therapy was immediately interrupted and antiacids and antiulcer therapies with pantoprazole IV 1 mg/kg BID, and sucralfate 40 mg/kg per os (OS) TID was instituted. Antibiotic therapy was continued but was switched to cefazolin 22 mg/kg IV TID and analgesic therapy with buprenorphine 10 μg/kg intramuscular (IM) TID was added. An antiemetic therapy with maropitant 1 mg/kg IV SID was added. Moreover, enteral feeding was instituted to satisfy his Rest Energy Requirement (RER) with a liquid enteral diet. After two days of therapy, the patient presented an onset of brown vomiting and his clinical conditions worsened. The dog became weak and nauseated, with pale mucous membranes and marked abdominal pain. Mild hypotension systolic arterial pressure (80 mmHg), tachycardia (180 b.p.m.) and hyperthermia 39.9 °C (103.82 °F) were noted. A lead II ECG showed sinus tachycardia. The peripheral packed cell volume, total plasma protein and lactate were 22% (RI 37–55%), 4.8 g/L (RI 5.5–7.7 g/L) and 6.7 mmol/L (RI < 2.5 mmol/L), respectively. A blood smear revealed neutrophilia with left shift and toxic changes. A repeated abdominal ultrasound showed an accumulation of abdominal effusion, which was sampled. A cytological exam suggested an aseptic neutrophilic exudate. However, a bacteriological sample was submitted for further evaluation. The abdominal effusion was biochemically analyzed and compared for glucose, creatinine, lactate and bilirubin concentrations with serum sample. Cavitary effusion showed a glucose concentration of 36 mg/dL, lactate level of 9.6 mmol/L, creatinine concentration of 1.3 mg/dL, a bilirubin concentration of 5.2 mg/dL and a potassium concentration of 5.0 mmol/L. Biochemical serum analysis showed a glucose concentration of 101 mg/dL (RI 74–143 mg/dL), lactate level of 6.7 mmol/L (RI < 2.5 mmol/L), creatinine level of 1.42 mg/dL (RI 0.5–1.8 mg/dL), bilirubin concentration of 0.3 mg/dL (RI 0.0–0.9 mg/dL) and a potassium concentration of 4.7 mmol/L (RI 3.5–5.0 mmol/L). Based on these findings, a biliary peritonitis was suspected. The owner was informed, and an explorative laparotomy was suggested. However, since a neoplastic condition was suspected, postoperative complications and a poor prognosis were notified to the owner, who decided to euthanize the patient humanely. To better understand the cause of the biliary peritonitis, a necropsy, with the owners’ consent, was performed. The median sagittal laparotomy was performed. At the opening of the abdominal cavity, a moderate amount of perivisceral-omental fat and the presence of abdominal brownish abdominal effusion was observed. Gastrointestinal tract inspection revealed the presence of perforating ulcer at the pylorus region level, and three other mucosal erosions were detected throughout the stomach (Figure 1). 

Abundant bile content was observed in the duodenum. The duodenal papilla was patent, and the common bile duct’s integrity was evaluated by inspecting the duodenal papilla and ascending the duct.

The regions of the colon-rectum presented blackish fecal material compatible with melena. The liver was slightly degenerate with distention of extrahepatic bile ducts. Gallbladder and common bile duct were intact. Additionally, no stones were observed in the gallbladder and biliary tree. Other extra abdominal compartments were unremarkable. Tissue samples from pathological organs were collected and fixed in 10% neutral buffered formalin for routine histological examination. Microscopically, the pyloric region showed multiple deep ulcers. Some of these were characterized by necrotic debris and neutrophils overlying the exposed connective tissue and fibrocellular smooth muscle (Figure 2A). Hemorrhage, fibrinous and neutrophilic exudates and necrotizing vasculitis were observed in the deep layers under the ulceration; neutrophilic ganglionitis of the myenteric plexus was observed (Figure 2B). 

One perforated ulcer was also observed. Severe fibrinous and neutrophilic peritonitis were observed in the omentum and serosa of the stomach, gallbladder, and the duodenum. In these organs, necrosis, fibrin deposition within and around the vessel wall (fibrinoid necrosis) and numerous neutrophils were observed in the serosal lining. Diffusely centrilobular to midzonal hepatocellular degeneration (fatty change) and necrosis with scant mononuclear infiltrates, and multifocal accumulation of hemosiderin in Kupffer cells were also detected in the liver. Mild lymphoplasmacytic enteritis was observed. In conclusion, the necroscopy showed severe biliary peritonitis due to gastric perforation without biliary abnormalities. 

## 3. Discussion

Biliary peritonitis is defined as an inflammatory condition due to free bile in the peritoneal cavity [5]. The presence of bile in the abdominal cavity is due to the spontaneous or iatrogenic (i.e., surgery) rupture of the biliary system. Cholangitis (primary or sequela to the gallbladder disease), obstruction of a bile duct, neoplasia of the biliary system, and trauma are the most frequent conditions causing the leakage of bile from the biliary system reported in the literature [6,7]. Canine cholangitis is often related to bacterial infection. *Escherichia coli* and *Enterococcus* spp. are the most isolated pathogens from bile. However, parasitic cholangitis is also reported (fluke-associated cholangitis). [8]. Moreover, systemic diseases, such as hyperadrenocorticism and hypothyroidism, have been demonstrated to be positively associated with biliary mucocele, a progressive chronic inflammatory process often found in dogs with cholangitis. Bile duct obstruction is known as extrahepatic biliary duct obstruction (EHBDO), a condition secondary to intramural, extramural, or intraluminal obstruction from various etiologies. Intramural obstructions occur when there is a thickening of the wall of the biliary tract with partial or complete closure of the duct. This condition generally is caused by severe chronic inflammatory processes. On the other hand, extramural obstructions are due to mass effects in organs adjacent to the biliary system causing compression onto the biliary ducts. Intraluminal obstructions occur when choledocholiths clog the bile duct lumen. Sequels of EHBDO are severe and vary from the derangements of hemostasis and hepatic injury to gallbladder wall necrosis and gallbladder rupture [9]. Gallbladder neoplasia is a rare and poorly documented cause of extrahepatic biliary disease in dogs and cats. The most commonly reported tumors are neuroendocrine carcinoma, followed by adenocarcinoma and leiomyoma. One report described gallbladder rupture as secondary to neuroendocrine carcinoma [10,11]. Bile constituents are toxic to tissue, causing alterations in vascular permeability and consequently, necrosis [12]. Altered vascular permeability promotes the exudation of fluid and translocation of endogenous anaerobic bacteria from the intestines to the liver, blood or into the peritoneum [12]. Furthermore, bile in the peritoneal cavity can affect host defense mechanisms against pathogens, such as bacteria, likely augmenting the susceptibility to sepsis and organ dysfunction [12,13]. Clinical signs are often vague and related to an acute abdomen. They could include depression, fever, anorexia, abdominal distention, icterus and, in general, signs due to SIRS (Systemic Inflammatory Response Syndrome) or shock [6]. The clinical signs typically start five days before, with only 71.3% of cases displaying clinical signs [14,15,16]. The short- and long-term survival for biliary surgery is 66%, with the worst outcome in those dogs requiring cholecystic-enterostomy [17]. According to a study conducted on 219 dogs with gallbladder mucocele (GBM), gallbladder rupture and bile peritonitis at the time of the surgery had a significantly higher risk of death than those that did not, and no association was found between survival and bacteriologic culture or antibiotic administration at the time of surgery [13]. A peri-operative mortality rate of 21.7–40% is reported for dogs undergoing cholecystectomy for GBM [13]. Most mortalities occur within the first two weeks after surgery [1]. The most common complications include bile peritonitis, sepsis, disseminated intravascular coagulation and surgical site-dehiscence [18]. In patients with effusion, it is essential to perform cytological and biochemical fluid analysis for a correct diagnosis. The abdominal effusion, in fact, is a non-specific finding that can result from several pathological processes [19]. To understand the underlying cause of the abdominal effusion, peripheral blood concentrations of various parameters are often compared to their abdominal counterparts. Different parameters could be taken into consideration, such as glucose and lactate in septic peritonitis [20,21,22], lipase in pancreatitis [22], triglycerides in chylous effusion, creatinine and potassium in uroabdomen [22,23,24], and bilirubin in bile peritonitis [22]. Bile peritonitis should be suspected when yellow-green or brown-orange abdominal fluid is obtained during an abdominocentesis [22]. In some instances, the fluid associated with bile peritonitis may also be reddish due to the large number of erythrocytes leaking from the altered permeability vascular system [22]. Cytologically, bile peritonitis effusions is characterized by greenish, gold, or black-brown pigments free in the background or within macrophages [25]. Sometimes it can be difficult to distinguish bile pigment from the pigment of red blood cell degradation products (hemosiderin) that appears as blue-green globular material; in this case, the evaluation of bilirubin concentration in the effusion may play a key role [26]. An abdominal fluid bilirubin concentration that is more than two times the serum bilirubin concentration confirms a diagnosis of bile peritonitis [6,26]. Interestingly, in the present case, no biliary lesion or rupture was found in the necropsy. Instead, a gastric perforation was identified, and the authors speculate that through this lesion, as a consequence of bilious vomiting, bile went into the abdominal cavity where it was sampled. Therefore, duodenogastric reflux (GDR) may have played a critical role in biliary peritonitis development. GDR is reported in both veterinary and human medicine, and it results from gastroduodenal dysmotility and changes in the speed of gastric emptying [27,28]. Gastrointestinal hormones, such as gastrin, cholecystokinin, and secretin, play a key role in reflux by influencing the secretion of gastric acids and regulating stomach motility [29]. The contents in the duodenum, such as bile, pancreatic and duodenal juice, retrograde to the stomach and lead to gastric inflammation. Bile acids and lysolecithin are the major components that destroy the barrier on the surface of gastric mucosa by dissolving phospholipids and cholesterol, which allows for back diffusion of hydrogen ions. As a consequence, gastritis, characterized by hyperemia, oedema, and erosion, occurs [30,31] and, histologically, foveolar hyperplasia occurs. Moreover, duodenal reflux liquids contain gut microbiome that can result in the imbalance of microbial flora in the stomach [32]. Moreover, the rise in the pH value due to alkaline bile creates a favorable environment for bacterial multiplication, further aggravating gastric inflammation [33]. In this patient, the probability of developing a gastric perforation could be facilitated by the administration of nonsteroidal anti-inflammatory drugs (NSAIDs), as reported in the dog’s anamnesis. Many scientific reports link gastroduodenal ulceration to the nonsteroidal anti-inflammatory drugs in several species, including dogs, cats and humans [34,35,36,37,38,39,40,41,42,43]. Gastric mucosal damage from NSAIDs can be from direct toxicity or from the inhibition of prostaglandin synthesis. Since COX-1 is constitutively expressed and COX-2 is an inducible enzyme, it was originally felt that inhibition of COX-1 and COX-2 was the main mechanism for the gastrointestinal side effects of NSAIDs [35,44,45]. This led to the development of NSAIDs with selective COX-2 inhibition. Studies looking at the safety of COX-2 selective NSAIDs in dogs have not shown a clear-cut advantage over non-specific COX-2 inhibitors [39,40,46,47]. Biases, such as the potential lack of statistical power in the veterinary studies, inaccurate assessment of the COX-2 specificity of the NSAIDs, and individual variability, may be the reasons behind this difference in dogs compared to humans [35,47,48]. In the present case, all erosions and gastric perforation were in the area adjacent to the pylorus. As previously reported, pylorus-antral region represents the most common site for NSAID-related ulcers in dogs [48]. Therefore, there was a reasonable correlation between the perforation site based on necroscopy and the NSAID administration.

## 4. Conclusions

This case report describes a case of biliary peritonitis related to gastric perforation. To the author’s best knowledge, this is the first case of biliary peritonitis in a dog associated with a gastric perforation, presumably secondary to the duodeno-gastric reflux associated with the administration of nonsteroidal anti-inflammatory drugs. This case report could represent a potential condition in patients suffering from gastric disorders and should be considered as a differential diagnosis of bile peritonitis.

## Figures and Tables

**Figure 1 vetsci-10-00384-f001:**
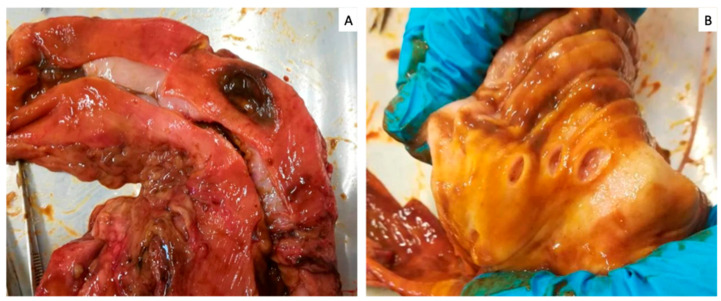
Stomach: (**A**) the gross appearance of perforating ulcer at the pylorus region and (**B**) three mucosal erosions.

**Figure 2 vetsci-10-00384-f002:**
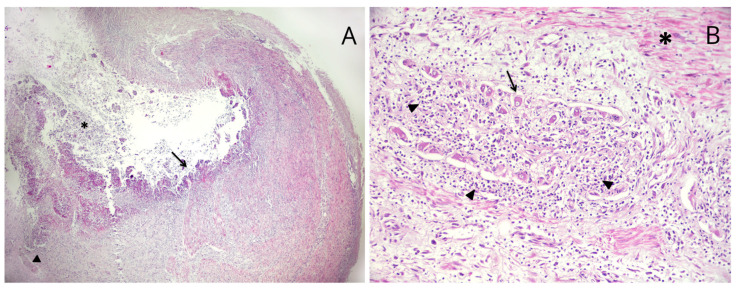
Microscopic appearance of the stomach. (**A**) Deep ulcer characterized by large lumen filled by necrotic debris (*). The ulcerated mucosa shows an exposure of mineralized connective tissue (arrow). Isolated vessel affected by vasculitis and thrombosis (arrowhead). (Hematoxylin and eosin, ×5). (**B**) Myenteric ganglia between internal circular muscular layer (*) and outer longitudinal muscular layer. Several myenteric neurons (arrow) appear to be surrounded by neutrophils (arrowhead) (neutrophilic ganglionitis). (Hematoxylin and eosin, ×20).

## Data Availability

Not applicable.

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
