# Peer review of "Unusual Case of Biliary Peritonitis in a Dog Secondary to a Gastric Perforation"

_vetsci, 2023, doi:10.3390/vetsci10060384_

Round 1

Reviewer 1 Report

The clinical report is interesting and is correctly described, reasoned and discussed.  The most relevant issues have been exposed:

1. Differential diagnosis: included gastric perforation

2. Side effects of NSAIDs: including COX-2 inhibitors

3. Diagnosis of bile peritonitis

4. Evolution of the patient: worsening hyperlactatemia

However, there are english language editing details that must be addressed for publication:

I would like to suggest the following changes:

Line 59: Doppler

Line 60: non-regenerative

Line 61: leukocytosis

Line 62: Venous gas analysis

Line 69: antacids

Line 77: Mild hypotension (80 mmHg) systolic or dyastolic arterial pressure?

Line 81: and toxic changes

Line 95: necropsy

Line 116: serosa

Line 133-134: organ dysfunction 

Comment for lines 62-64: the authors documented hyperlactatemia but do not provide base excess (or deficit) or bicarbonate concentration.  I believe that "the patient showed hyperlactatemia that may cause or shall induce metabolic acidosis" better than "showed mild hyperlactatemia with a concurrent metabolic acidosis" 

Author Response

The clinical report is interesting and is correctly described, reasoned and discussed.  The most relevant issues have been exposed:

Question 1. Differential diagnosis: included gastric perforation

Response 1: Thank the reviewer for the comment. We do not include gastric perforation in differential diagnosis because it is not reported in bibliography. The most interesting aspect of this case report is the biliary peritonitis due to gastric perforation never reported in literature.

Question 2. Side effects of NSAIDs: including COX-2 inhibitors

Response 2: Modified in line 182.

Question 3. Diagnosis of bile peritonitis

Response 3: Modified.

Question 4: Evolution of the patient: worsening hyperlactatemia

Response 4: Modified.

Question 5: Line 59: Doppler

Response 5: Thank the reviewer for the comment. Modified.

Question 6: Line 60: non-regenerative

Response 6: Thank the reviewer for the comment. Modified.

Question 7: Line 61: leukocytosis

Response 7: Thank the reviewer for the comment. Modified.

Question 8: Line 62: Venous gas analysis

Response 8: Thank the reviewer for the comment. Modified.

Question 9: Line 69: antacids

Response 9: Thank the reviewer for the comment. Modified.

Question 10: Line 77: Mild hypotension (80 mmHg) systolic or dyastolic arterial pressure?

Response 10: Thank the reviewer for the comment. Modified.

Question 11: Line 81: and toxic changes

Response 11: Thank the reviewer for the comment. Modified.

Question 12: Line 95: necropsy

Response 12: Thank the reviewer for the comment. Modified.

Question 13: Line 116: serosa

Response 13: Thank the reviewer for the comment. Modified.

Question 14: Line 133-134: organ dysfunction 

Response 14: Thank the reviewer for the comment. Modified.

Question 15: Comment for lines 62-64: the authors documented hyperlactatemia but do not provide base excess (or deficit) or bicarbonate concentration.  I believe that "the patient showed hyperlactatemia that may cause or shall induce metabolic acidosis" better than "showed mild hyperlactatemia with a concurrent metabolic acidosis".

Response 15: Thank the reviewer for the comment. We modified line 64 adding the bicarbonate concentration. Please see line 62-64.

Reviewer 2 Report

This is an interesting case report. 

Major issues in the paper for me include

 - Bobtail neutered male dog Line 50; 

Why is bobtail significant here? what is the breed?

- elective splenectomy and castration performed 3 days before 51 due to the presence of a splenic and testicle neoplasia

This is a huge piece of information missing regarding neoplasia. What was the splenic and testicle neoplasia? I don't think splenectomy in any case is elective in veterinary medicine.  Was there any possibility of iatrogenic trauma to the biliary tree. Was the gall bladder evaluated for resealing? Tears can be small and can be missed on gross examination. 

- Venous emogas analysis 62

gas?

- mild hyperlactatemia with a concurrent metabolic acidosis 3,8 mmol/L (RI < 2,5 63; 

What is mild hyperlactatemia? Where is the RI.

- Did someone consider the possibility that most of the damage was done by gastric acid versus the bile?

- I would think that the actual amount of bile that leaked would not be significant. Was the gall bladder full during ultrasound and during the necropsy exam?

- Was there any pancreatitis?

- Was bile acid concentrations considered?

The report should be rewritten. 

Author Response

Question 1: Bobtail neutered male dog Line 50; Why is bobtail significant here? what is the breed?

Response 1: Thank the reviewer for the comment. The Bobtail is a specifical dog breed also known as or Old English Sheepdog. We modified with “ Bobtail (or Old English Sheepdog)”, please see line 50.

Question 2: elective splenectomy and castration performed 3 days before 51 due to the presence of a splenic and testicle neoplasia. This is a huge piece of information missing regarding neoplasia. What was the splenic and testicle neoplasia? I don't think splenectomy in any case is elective in veterinary medicine.  Was there any possibility of iatrogenic trauma to the biliary tree. Was the gall bladder evaluated for resealing? Tears can be small and can be missed on gross examination. 

Response 2: Thank the reviewer for the comment. The dog was presented to our veterinary center by the referent 3 days after splenectomy and removal of the testicles. The histological examination had not yet been reported due to the technical times for the examination. We modified the line 51 with “after splenectomy” and not “after an elective splenectomy”. The biliary tree was evaluated during necropsy exam. We add this part in line 105 with “the integrity of the common bile duct was performed by inspecting the duodenal papilla and ascending the duct”.

Question 3: Venous emogas analysis 62

gas?

Response 3: Modified.

Question 4: mild hyperlactatemia with a concurrent metabolic acidosis 3,8 mmol/L (RI < 2,5 63; 

What is mild hyperlactatemia? Where is the RI.

Response 4: Thank the reviewer for the comment. According with Boag et al. “ Mild hypoperfusion seems to be associated with plasma lactate concentrations of 3–5 mmol/L (27-45 mg/dL), moderate with 5–7 mmol/L (45-63 mg/dL), and severe with > 7 mmol/L (> 63 mg/dL). However we modified with “hyperlactatemia”.Ref. Boag AK, Hughes D. Assessment and treatment of perfusion abnormalities in the emergency patient. Vet Clin North Am Small Anim Pract. 2005 Mar;35(2):319-42.

Question 5: Did someone consider the possibility that most of the damage was done by gastric acid versus the bile?

Response 5: The reviewer is right. Gastric acid can worsen peritonitis but, in this case, the quantified datum was the presence of bile which, as explained in the discussions, causes severe peritonitis.

Question 6: I would think that the actual amount of bile that leaked would not be significant. Was the gall bladder full during ultrasound and during the necropsy exam?

Response 6: Thank the reviewer for the question. As reported from line 108, the necropsy exam showed that Gallbladder and common bile duct were intact. Cavitary effusion showed a bilirubin concentration of 5,2 mg/dl and serum bilirubin concentration of 0,3 mg/dl. Based on these findings was suspected a biliary peritonitis. According with the literature, abdominal fluid bilirubin concentration that is more than two times concurrent serum bilirubin concentration confirms a diagnosis of bile peritonitis.

Question 7: Was there any pancreatitis?

Response 7: Thank the reviewer for the question. Sonography exam didn’t show alteration in pancreatic area. We didn’t perform a specific test for Canine pancreatic lipase.

Question 8: Was bile acid concentrations considered?

Response 8: Thank the reviewer for the comment. As reported in the manuscript, the abdominal effusion was biochemical analyzed and compared for glucose, creatinine, lactate and bilirubin concentrations with serum sample. Please line 85.

Reviewer 3 Report

Well, this is a clinical case that has the originality that no other described in the bibliography has been found, however I have some considerations that I would like to be clarified.

First of all, in line 67, they comment that to reduce stomach contents they place a nasogastric feeding tube. Could you explain to me why they have chosen this system and not the placement of a tube for oral lavage under anesthesia? In addition to avoid possible aspiration pneumonia as a complication when performing said procedure without much control?

In line 68 they comment that on suspicion of presenting a gastric ulcer, they treat him with antacid therapy, which in principle does not pose any problem from my point of view, but in this case, and given the worsening of the clinical signs, what Isn't performing a diagnostic endoscopy inevitable, especially when I already had a lactate of 3.8? Also, could you explain to me why they change antibiotic therapy and why they administer buprenorphine intramuscularly if he is supposed to have an intravenous catheter?

In line 81, when they comment that the abdominal ultrasonography was given, they did not comment on any finding except the accumulation of abdominal effusion. Could you give any details on how the rest of the organs were visualized, mainly the gallbladder and ducts?    

On line 93, why do they say they suspect a neoplasia? What findings do you have to think about that possibility?

In the conclusions, in my opinion, omit the word Bobtail (line 194), since the race of the patient is not relevant in the description of the clinical case.

Author Response

Question 1: First of all, in line 67, they comment that to reduce stomach contents they place a nasogastric feeding tube. Could you explain to me why they have chosen this system and not the placement of a tube for oral lavage under anesthesia? In addition to avoid possible aspiration pneumonia as a complication when performing said procedure without much control?

Response 1: Thank the reviewer for the comment. A nasogastric feeding tube was placed because it is less invasive and only requires a local anesthesia procedure compared to gastric lavage. Furthermore, the placement of nasogastric feeding tube allows the management of the patient during hospitalization.

Question 2: In line 68 they comment that on suspicion of presenting a gastric ulcer, they treat him with antacid therapy, which in principle does not pose any problem from my point of view, but in this case, and given the worsening of the clinical signs, what Isn't performing a diagnostic endoscopy inevitable, especially when I already had a lactate of 3.8? Also, could you explain to me why they change antibiotic therapy and why they administer buprenorphine intramuscularly if he is supposed to have an intravenous catheter?

Response 2: Thank the reviewer for the comment. A diagnostic endoscopy was not performed because the owner decided to humanely euthanize the patient due to the operative complications and the poor prognosis.

Question 3: In line 81, when they comment that the abdominal ultrasonography was given, they did not comment on any finding except the accumulation of abdominal effusion. Could you give any details on how the rest of the organs were visualized, mainly the gallbladder and ducts?    

Response 3: Modified.

Question 4: On line 93, why do they say they suspect a neoplasia? What findings do you have to think about that possibility?

Question 4: Thank the reviewer for the comment. The histological report of the splenic and testicular neoplasia is missing because the patient was sent to our veterinary center by the referent. Furthermore, the technical processing times of the histological examination did not allow for a diagnosis before the patient's euthanasia.

Question 5: In the conclusions, in my opinion, omit the word Bobtail (line 194), since the race of the patient is not relevant in the description of the clinical case.

Response 5: Thank the reviewer for the comment. Modified. Please see line 197.

Reviewer 4 Report

Comments on VetSci-2348437-V1, entitled “Unusual case of biliary peritonitis in a dog secondary to a gastric perforation.” by Pavone et al.

The manuscript reports an interesting and well documented clinical case, and images are very illustrative. There are some minor changes that should be amended, and the English should be reviewed by an English native person.

Major Changes

Lines 130-131- Please clarify the sentence “Altered vascular permeability promotes transudation of fluid and translocation of endogenous anaerobic bacteria from the liver, intestines, and blood into the peritoneum.”- Are authors sure that there are usually anaerobic bacteria living in the liver that translocate into the peritoneum??? Or authors mean that anaerobic bacteria translocate from the intestine into the liver, blood and peritoneum?? Please clarify.

Please also confirm the following: if there is an increased vascular permeability, it promotes exudation (inflammatory oedema) not transudation (non-inflammatory oedema). 

Minor changes

Line 52- Replace the sentence” Two days after surgery dog showed” to “Two days after surgery the dog showed …”

Line 60- Please amend “non-rigenerative”.

Line 60- Amend “…anemia red cells…” to “…anemia, red cells…”

Line 61-Write “hematocrit” and “hemoglobin” in full before Hct and Hbg, respectively. 

Line 79-Write “PCV” in full. 

Line 88- Rewrite the sentence “Biochemical serum analysis was glucose concentration of 101 mg/dl (RI 74-143 mg/dl), lactate level of 6,7 mmol/l (RI <2,5 mmol/L)…” to “Biochemical serum analysis showed (or revealed, as authors prefer) a glucose concentration of 101 mg/dl (RI 74-143 mg/dl), lactate level of 6,7 mmol/l (RI <2,5 mmol/L)…

Line 91- Rewrite the sentence: “Based on these findings was suspected a biliary peritonitis” to “Based on these findings, a biliary peritonitis was suspected”.

Line 97- Replace “…revealed presence of perforating…” to “…revealed the presence of a perforating…” 

Line 110- Replace “Microscopically, pyloric region showed…” to “Microscopically, the pyloric region showed…

Line 111- Amend “…characterized by floor of necrotic debris...” to  “characterized by necrotic debris

Line 116- Amend “sierosa

Line 142- Please amend …”administration to the time of surgery” to “administration at the time of surgery

Lines 150-151- Please amend “…abdominal effusion peripheral blood concentrations of various parameters are often compared to their abdominal counterparts…” to “…abdominal effusion, peripheral blood parameters are often compared to their abdominal counterparts

References: There are some spaces missing. Sometimes after the journal authors use a dott and sometimes use a semicolon. Please uniform the criteria.

English should be reviewed by an English native person.

Author Response

Question 1: Lines 130-131- Please clarify the sentence “Altered vascular permeability promotes transudation of fluid and translocation of endogenous anaerobic bacteria from the liver, intestines, and blood into the peritoneum.”- Are authors sure that there are usually anaerobic bacteria living in the liver that translocate into the peritoneum??? Or authors mean that anaerobic bacteria translocate from the intestine into the liver, blood and peritoneum?? Please clarify.

Please also confirm the following: if there is an increased vascular permeability, it promotes exudation (inflammatory oedema) not transudation (non-inflammatory oedema). 

Response 1: Thank the reviewer for the comment. Modified. Please see line 145-146

Minor changes

Question 2: Line 52- Replace the sentence” Two days after surgery dog showed” to “Two days after surgery the dog showed …”

Response 2: Modified. Please see line 90.

Question 3: Line 60- Please amend “non-rigenerative”.

Response 3: Modified.

Question 4: Line 60- Amend “…anemia red cells…” to “…anemia, red cells…”

Response 4: Modified.

Question 5: Line 61-Write “hematocrit” and “hemoglobin” in full before Hct and Hbg, respectively. 

Response 5: Modified.

Question  6: Line 79-Write “PCV” in full. 

Response 6: Modified.

Question  7: Line 88- Rewrite the sentence “Biochemical serum analysis was glucose concentration of 101 mg/dl (RI 74-143 mg/dl), lactate level of 6,7 mmol/l (RI <2,5 mmol/L)…” to “Biochemical serum analysis showed (or revealed, as authors prefer) a glucose concentration of 101 mg/dl (RI 74-143 mg/dl), lactate level of 6,7 mmol/l (RI <2,5 mmol/L)…

Response 7: Modified.

Question 8: Line 91- Rewrite the sentence: “Based on these findings was suspected a biliary peritonitis” to “Based on these findings, a biliary peritonitis was suspected”.

Response 8: Modified.

Question 9: Line 97- Replace “…revealed presence of perforating…” to “…revealed the presence of a perforating…” 

Response 9: Modified.

Question 10: Line 110- Replace “Microscopically, pyloric region showed…” to “Microscopically, the pyloric region showed…

Response 10: Modified.

Question 11: Line 111- Amend “…characterized by floor of necrotic debris...” to  “characterized by necrotic debris

Response 11: Modified.

Question 12: Line 116- Amend “sierosa

Response 12: Modified.

Question 13: Line 142- Please amend …”administration to the time of surgery” to “administration at the time of surgery

Response 13: Modified.

Question 14: Lines 150-151- Please amend “…abdominal effusion peripheral blood concentrations of various parameters are often compared to their abdominal counterparts…” to “…abdominal effusion, peripheral blood parameters are often compared to their abdominal counterparts

Response 14: Modified.

Question 15: References: There are some spaces missing. Sometimes after the journal authors use a dott and sometimes use a semicolon. Please uniform the criteria.

Response 15: Modified.

Round 2

Reviewer 2 Report

I think the changes are satisfactory 

Should be edited by a language editor